# Use of corticoids and non-steroidal anti-inflammatories in the treatment of rheumatoid arthritis: Systematic review and network meta-analysis

**Mariana Del Grossi Paglia**[¤]*, **Marcus Tolentino Silva**[☯], **Luciane Cruz Lopes**[☯], **Silvio Barberato-Filho**[‡], **Lauren Giustti Mazzei**[‡], **Flavia Casale Abe**[‡], **Cristiane de Cássia Bergamaschi**[☯]

Graduate Program in Pharmaceutical Sciences, University of Sorocaba, Sorocaba, State of São Paulo, Brazil

☯ These authors contributed equally to this work.
¤ Current address: Department of Pharmaceutical Sciences, University of Sorocaba, Sorocaba, State of São Paulo, Brazil
‡ These authors also contributed equally to this work.
* maridelgrossi@gmail.com

**Data Availability Statement:** All relevant data are within the paper and its Supporting Information files.

## Abstract

Evidence on the use of non-steroidal anti-inflammatory drugs (NSAIDs) and corticoids for rheumatoid arthritis (RA) is inconclusive and is not up to date. This systematic review assessed the effectiveness and safety of these anti-inflammatories (AI) in the treatment of RA. COCHRANE (CENTRAL), MEDLINE, EMBASE, CINAHL, Web of Science and Virtual Health Library were searched to identify randomized controlled trials (RCT) with adults which used AI (dose represented in mg/day) compared with placebo or active controls and was carried out up to December of 2019. Reviewers, in pairs and independently, selected studies, performed the data extraction and assessed the risk of bias. The quality of the evidence was assessed by GRADE. Network meta-analyses were performed using the Stata v.14.2. Twenty-six articles were selected (NSAIDs = 21 and corticoids = 5). Naproxen 1,000 improved physical function, reduced pain and the number of painful joints compared to placebo. Etoricoxib 90 reduced the number of painful joints compared to placebo. Naproxen 750 reduced the number of swollen joints, except for etoricoxib 90. Naproxen 1,000, etoricoxib 90 and diclofenac 150 were better than placebo regarding patient assessment. Assessment physician showed that NSAIDs were better than placebo. Meta-analyses were not performed for prednisolone and prednisone. Naproxen 1,000 was the most effective drug and celecoxib 200 showed fewer adverse events. However, the low quality of the evidence observed for the outcomes with NSAIDs, the absence of meta-analyses to assess the outcomes with corticoids, as well as the risk of bias observed, indicate that future RCT can confirm such findings.

**Funding:** This project is funded by governmental Program Graduate Education Institutions—PROSUP—CAPES/UNISO".

**Competing interests:** The authors have declared that no competing interests exist.

## Introduction

Rheumatoid arthritis is a chronic and progressive systemic inflammatory disease. Its prevalence is about 5 in every 1,000 individuals [1], occurring often during their most productive years. It affects twice as many women than men [2, 3]. Patients suffering from rheumatoid arthritis usually require analgesic and anti-inflammatory drugs to control disease symptoms [4, 5].

Non-steroidal anti-inflammatory drugs (NSAIDs) and corticoids (steroid anti-inflammatory drugs) are commonly used in patients as adjuvants to rheumatoid arthritis treatment, as they can promote benefits by reducing pain and inflammation caused by the disease [5–8].

Disease-modifying antirheumatic drugs (DMARDs) in combination with corticoids can be used as first choice therapy options and their administration should be started as soon as the diagnosis of rheumatoid arthritis has been confirmed, in an effort to achieve remission and prevent the increase of disease activity [4, 5, 9, 10].

DMARDs administered in combination with prednisone or prednisolone at lower doses ($\leq$10 mg/day) and for short periods (<3 months) can help reduce symptoms and radiographic progression [11–17]. NSAIDs can also be prescribed for symptomatic control while the effects of synthetic or biological DMARDs take place, usually at the lowest dose for the shortest possible period [18, 19]. NSAIDs recommended as first choices include ibuprofen, naproxen, potassium diclofenac and sodium diclofenac for patients with mild, moderate or high disease activity [6, 20].

Although there are systematic reviews evaluating the use of NSAIDs [6, 20] and corticoids [21–23] for the treatment of rheumatoid arthritis, no up-to-date evidence was found on this topic and these drugs are routinely used by patients suffering from this condition. Network meta-analysis can be a strategy for dealing with existing evidence. Published studies on this topic are old and safety data on long-term use are very unclear [6]. Thus, this study performed a systematic review of randomized clinical trials (RCT) on the effectiveness and safety of use of these anti-inflammatories in the treatment of rheumatoid arthritis.

## Methods

The systematic review was performed according to the recommendations specified in the Cochrane Handbook for Interventional Reviews [24, 25] and reported according to the Preferred Reporting Items for Systematic Reviews and Meta-Analysis (PRISMA) extension for network meta-analysis [26].

### Protocol and registration

The review protocol was registered by the International Prospective Register of Systematic Reviews (PROSPERO, protocol number CRD42017073532) and was previously published [27].

### Differences between protocol and review

Some adjustments to the protocol version were made such as: i) eligibility criteria was altered to include RCT in which patients were diagnosed with rheumatoid arthritis regardless of diagnostic criteria, ii) trials of cross-over design were excluded due to the difficulty of using data to perform comparisons of results; iii) some adjustments were necessary in the search strategy; iv) we performed the indirect meta-analyzes due to the heterogeneity presented by RCT included in this review; and v) subgroup analyzes were not performed due to heterogeneity between studies.

## Eligibility criteria

**Types of studies.**    RCT that compared NSAIDs or corticoids to another therapy (placebo or active control) for rheumatoid arthritis were considered eligible. Studies where only the abstract was available or if they had fewer than 200 participants or trials of cross-over design were excluded.

**Types of participants.**    Studies involving adults (≥18 years old) diagnosed with rheumatoid arthritis were considered eligible. Studies in which more than 20% of the patients suffered from another inflammatory disease were excluded, except in cases where results for the studied population could be separated from other analyses.

**Types of interventions.**    Experimental group: NSAIDs (aceclofenac, aspirin, bufexamac, diclofenac, etodolac, fenclofenac, fenoprofen, flurbiprofen, ibuprofen, indomethacin, ketoprofen, ketorolac, meclofenamic acid, mefenamic acid, naproxen, niflumic acid, oxaprozin, oxyphenbutazone, phenylbutazone, piroxicam, sulindac, suprofen, tenoxicam, tiaprofenic acid, tolfenamic acid, nabumetone, meloxicam, celecoxib and etoricoxib) and SAIDs/corticoids (beclomethasone, betamethasone, budesonide, dexamethasone, flunisolide, fluticasone, fludrocortisone, hydrocortisone, methylprednisolone, prednisolone, prednisone and triamcinolone) at any dose, duration and route of administration and that are commercially available;

Control group: placebo or any active control.

**Types of outcome measures.**    Primary outcomes measure: pain (Visual Analogue Scale–VAS, patient global impression or other scale); physical function (measured using the Health Assessment Questionnaire–HAQ or a modified HAQ) [28]; number of swollen joints; number of painful joints; morning stiffness (time in minutes or hours); grip strength (indicator of general strength and general health); patients' and physicians' global assessment, disease progression as assessed based on radiological imaging of joints; quality of life (Short Form-36 and other scales).

Secondary outcomes measures included: adverse events and serious adverse events-SAE (such as death, life-threatening events, hospitalization, disability or permanent damage); withdrawal from the study; satisfaction with current treatments and consumption of rescue medication.

## Information sources

We searched the following electronic databases with no restrictions regarding publication status or language: COCHRANE (CENTRAL), MEDLINE; EMBASE; CINAHL; Web of Science and Virtual Health Library. References for all included studies, other reviews, guidelines and related articles were searched examining reference lists. Ongoing studies were searched in the trial registry ClinicalTrials.gov and World Health Organization International Clinical Trials Registry Platform. The searching was carried out in order to identify all relevant publications up to December 14 of 2019.

## Search

The search strategy was created using terms of the Medical Subject Headings (MeSH) and keywords. The associated keywords: i) intervention (anti-inflammatory agents); ii) condition (rheumatoid arthritis), and iii) methodological filters were applied to limit retrieval to RCT.

The search strategy was adapted for each database and designed with the assistance of a trained librarian (S1 File).

## Study selection

Four reviewers (MDGP, SB-F, LGM, FCA), working in pairs and independently, screened titles and abstracts. The same reviewers, in pairs and independently, assessed eligibility of each

full-text article. In case of duplicate publications, we would just include the article with most complete data, however this situation did not occur. Disagreements were resolved by consensus or by a third review author (CCB or LCL) if necessary.

## Data collection process

All reviewers, in pairs and independently, extracted the data using standardized and pretested forms with instructions and contacted study authors to clarify any uncertainties. Studies where important data were incomplete or missing, we contacted the authors, however, we have not received reply from any authors. Whenever possible, we computed missing standard deviation (SD) from other statistics, such as standard error (SE) [24]. For the studies that did not provide enough data, we verified whether these values could be extracted from graphs using web based tools (https://automeris.io/WebPlotDigitizer/).

The information extracted were: year and country of the publication, register protocol, study design, characteristics of the population (diagnostic criteria, pain relief medications, number of patients, mean and standard deviation age, percentage of women); interventions and comparators (drug, dose diary, via of administration, duration of the treatment in weeks), risk of bias and outcomes.

## Geometry of the network

The data were summarized in a network meta-analysis. The model was proposed by Bucher et al. [29] and draws on both direct evidence (treatments compared in the same trial) and indirect evidence (different treatments studied in separate trials, but compared when they use a common comparator), with the benefit of randomization in each study retained.

Network meta-analysis using mixed treatment comparisons technique was carried out to unite in a single analysis direct and indirect evidences, the main objective being increasing precision of the estimation. The network diagram is made up of lines and nodes. In the diagram, the notes represent every intervention, and the size of the nodes means the number of participants. The lines indicate direct comparisons between different interventions and the thickness of the line means the amount of studies [30].

## Risk of bias within individual studies

Using a modified version of the Cochrane collaboration risk of bias tool [24], the same reviewers assessed the risk of bias for each trial, in pairs and independently, according to the following criteria: random sequence; allocation concealment; blinding of the patients, care provider and outcome assessor for each outcome measure; incomplete outcome data; selective outcome reporting; and other biases.

To determine the risk of bias of a study, each criterion was rated as 'definitely yes', 'probably yes', being assigned a low risk of bias and 'probably no' and 'definitely no', assigned a high risk of bias [31]. Disagreements were resolved by consensus.

Incomplete outcome data, lost follow-up less than 10% and a difference of less than 5% in missing data in intervention and control groups were considered low risk of bias. In order to determine whether there was reporting bias or not, we first determined whether the protocol for the assessed RCT was published before recruitment of patients had started. For studies published after July 1st, 2005, we screened the Clinical Trial Register at the International Clinical Trials Registry Platform of the World Health Organization (http://apps.who.int/trialsearch) [32].

In cases where study protocol registration reports and safety results were not found, we used the classification for high risk of bias. The absence of a criterion for diagnosis of

rheumatoid arthritis was identified as a possible source of bias and classified as high risk of bias in the criterion "other risks of bias".

The bias classification was done using the Review Manager 5 software and a third review author (CCB or MTS) carried out any final decisions when necessary.

## Summary measures and methods of analyses

Analyses were carried out for each anti-inflammatory drug and for each outcome of interest. Estimates of comparative effectiveness were measured using standardized mean differences (SMD) with associated 95% confidence intervals (95% CI); and estimates of comparative safety were measured using odds ratio (OR) with 95% CI. Subgroup analysis could not be performed due to heterogeneity of the studies. The graphical representation of meta-analyses was not performed since there were few direct comparisons.

We are provided summary tables when the meta-analysis was not appropriate (S2 File). The analyses were carried out using Stata software (version 14.2). We adopted a comparison of mixed treatment with mixed generalized linear models to analyze the indirect and direct comparisons between the networks. The comparisons presented were derived from indirect and direct comparison, if available. The maximum restricted likelihood method was used to estimate the random effect model.

The inconsistency was evaluated using the loop-specific approach to evaluate the agreement between direct and indirect estimates, and it was regarded as a better consistency when the inconsistency factors (IF) was equal to zero [33]. We assess the possibility of intransitivity comparing trials participants between direct comparisons that contributed to an indirect comparisons [34]. Publication bias was graphically accessed via a comparison-adjusted funnel plot, and Egger's test were applied to measure the asymmetry; the results of Egger test (P > .05) were defined as non-significant publication bias among included studies [35].

We calculated the relative ranking of agents for induction of clinical remission as their surface under the cumulative ranking (SUCRA), which represents the percentage of efficacy or safety achieved by a drug compared to other that is always the best without uncertainty (SUCRA = 100%) [36]. This parameter was used to estimate the average treatment ranking to explore potential orderings of treatments. The trial nodes that were not connected to the network were excluded.

## Quality of evidence

We followed the Grading of Recommendations Assessment, Development and Evaluation (GRADE) approach to appraise the confidence in estimates derived from network meta-analysis of outcomes [34, 37]. RCT start at high confidence and can be rated down based on risk of bias, indirectness, imprecision, inconsistency and publication bias; they can then be graded at levels of moderate, low and very low confidence [38].

The risk of bias was evaluated for each outcome as low, moderate or high and is represented in figures of network meta-analysis in colors green, yellow and red, respectively. If direct and indirect estimates were coherent, then the higher of their ratings was assigned to the network meta-analysis estimates (S3 File).

## Results

### Study selection

Of a total of 10,498 publications (reasons for exclusion are in S4 File), 26 studies met the inclusion criteria (NSAIDs = 21 and corticoids = 5) (Fig 1, S5 File).

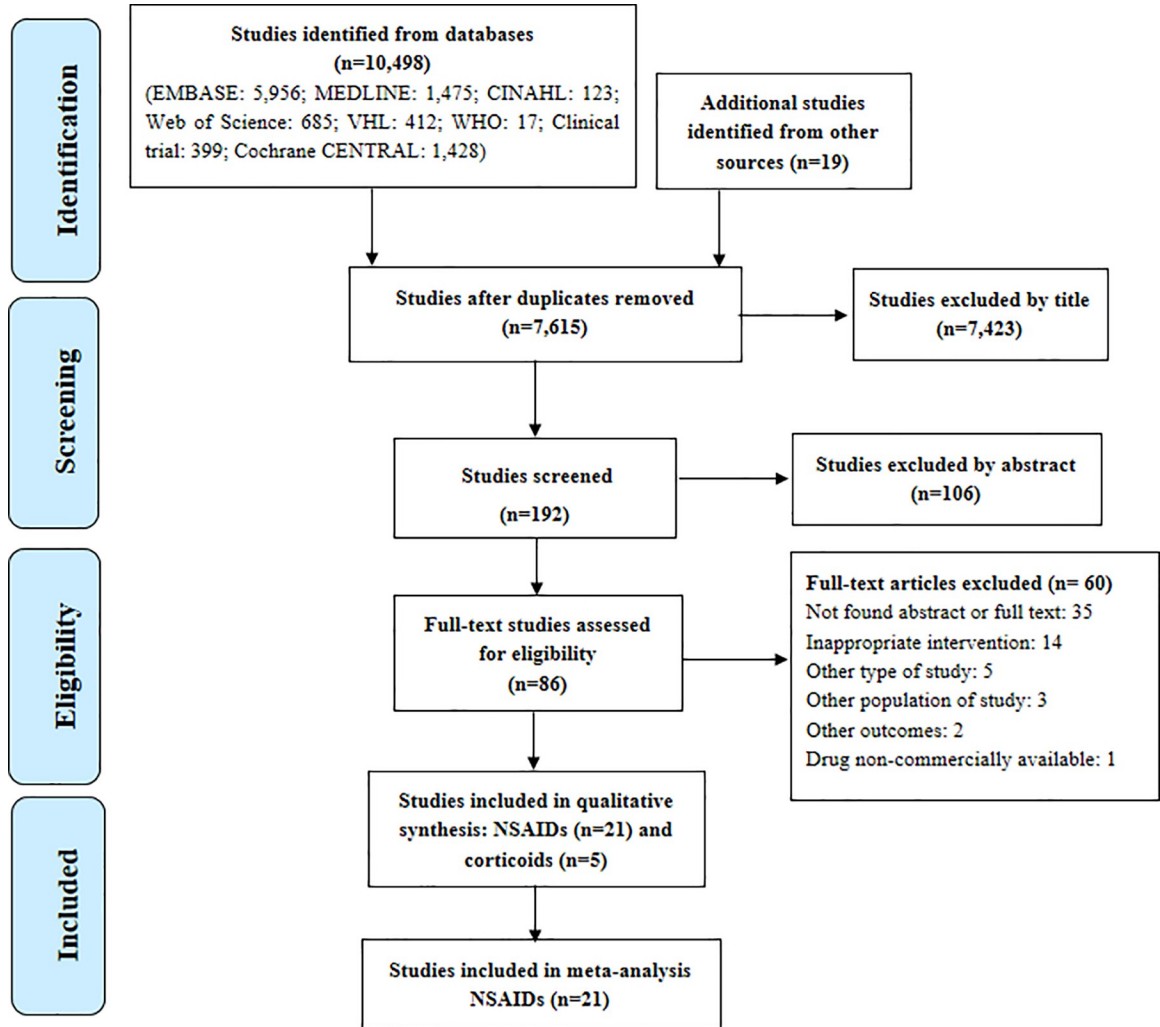

**Fig 1. Study flowchart.** VHL: Virtual Health Library; WHO: World Health Organization; NSAIDs: Non-steroidal anti-inflammatory drugs; corticoids drugs.

### Non-steroidal anti-inflammatories for rheumatoid arthritis

**Description of studies.** Twenty-one trials (10,503 patients) were included in this review. Most studies reported the functional class of the disease (n = 17). Clinical trials comprised ten NSAIDs (aceclofenac, aspirin, celecoxib, diclofenac, etodolac, etoricoxib, indomethacin, keto-profen, meloxicam, nabumetone, naproxen, piroxicam and tenoxicam). One study evaluated the use of a patch formulation, the others described oral administration. Follow-up time ranged from 14 to 182 days. Most studies reported the use of rescue medication (n = 16) and 17 studies described concomitant use of DMARD therapy, but did not describe the medicine used. The mean age of the patients varied between 46.9 and 58.7 years (Table 1).

**Risk of bias (Fig 2).** One study had minimum risk of bias [39]. Allocation concealment was insufficiently described in the majority them [40–55].

Shi et al. (2004) [53] did not describe the blinding of patients or healthcare professionals due to the fact the study was an open trial in which both researchers and participants were aware of which treatments were administered.

**Table 1. Characteristics of the included studies on non-steroidal anti-inflammatories for rheumatoid arthritis (n = 21).**

| Study | Functional capacity | Interventions (dose in mg/day) | Outcomes reported | Sample size (N) | Lost follow-up (%) | DMARDs Use | Rescue medication | Duration (weeks) | Mean age (years) | Woman (%) |
|---|---|---|---|---|---|---|---|---|---|---|
| **Bernhard et al., 1987[56]** | II or III | nabumetone 1,000, aspirin 900 | 5,6,7,8, 10 | 234 | 49.1 | not allowed | acetaminophen | 24 | 50.7 | 75 |
| **Collantes et al., 2002[40]** | I, II or III | placebo, etoricoxib 90 naproxen 1,000 | 1,3,4,7,8,9,10 | 687 | 29.7 | allowed | aspirin | 12 | 52.3 | NR |
| **Emery et al., 1992 [41]** | I, II or III | nabumetone 2,000 naproxen 1,000 | 1,5,10 | 284 | 4.9 | allowed | acetaminophen | 12 | 53.2 | NR |
| **Emery et al., 1999 [59]** | I, II or III | celecoxib 400 diclofenac 150 | 1,2,3,4,5,7,8,10 | 497 | 31.8 | allowed | NR | 24 | 55.2 | 96.7 |
| **Furst et al., 2002 [39]** | NR | placebo, meloxicam 7.5, 15, 22.5 diclofenac 150 | 1,2,3,4,7,8, 10 | 888 | 0.7 | allowed | acetaminophen | 12 | 55.4 | NR |
| **Geusens et al., 2002[42]** | I, II or III | Placebo naproxen 1,000 | 3,4,5,7,8,10 | 1023 | NR | allowed | acetaminophen | 12 | 53.6 | 82.8 |
| **Geusens et al., 2004[43]** | I, II or III | naproxen 500 placebo | 1,3,4,7,8,9,10 | 726 | 54.9 | allowed | acetaminophen | 26 | 53.5 | 88 |
| **Gibofsky et al., 2007[44]** | II or III | naproxen 1,000 placebo | 1,2,3,4,5,7,8,9,10 | 340 | 49.4 | allowed | acetaminophen | 12 | 55.9 | 68.5 |
| **Jacob et al., 1986 [45]** | I, II or III | placebo, etodolac 50, 100, 200, aspirin 3,900 | 1,3,4,5,6,10 | 264 | 42.4 | allowed | acetaminophen | 6 | 52.9 | 60.2 |
| **Kawai et al., 2010 [46]** | I, II, III or IV | placebo, ketoprofen 20 | 1,10 | 652 | 3.7 | allowed | NR | 2 | 58.7 | 85.8 |
| **Kornasoff et al., 1996[47]** | I, II or III | aceclofenac 200 indomethacin 100 | 3,4,5,6,7,8,10 | 219 | 17.8 | NR | acetaminophen | 12 | 56.0 | 70.7 |
| **Krug et al., 2000 [48]** | I, II or III | nabumetone 2,000 naproxen 1,000 | 3,4,7,8,10 | 344 | 0.6 | allowed | acetaminophen | 12 | 54.0 | 70.9 |
| **Lightfoot, 1997 [49]** | NR | etodolac 400, 600 piroxicam 20 | 3,4,5,10 | 361 | 37.3 | allowed | acetaminophen | 12 | 57.0 | 84.2 |
| **Matsumoto et al., 2002[50]** | I, II or III | placebo, etoricoxib 90 naproxen 1,000 | 1,3,4,7,8,9,10 | 448 | 68.7 |  | aspirin | 12 | 55.6 | NR |
| **Pasero et al., 1995 [51]** | NR | aceclofenac 200 diclofenac 150 | 1,5,6,10 | 327 | 7.6 | NR | NR | 24 | 50.7 | 81.3 |
| **Perez Ruiz; Alonso Ruiz; Ansoleaga, 1996 [52]** | NR | aceclofenac 200 tenoxicam 20 | 1,5,6,10 | 237 | 13 | not allowed | acetaminophen | 12 | 56.6 | 98.7 |
| **Shi et al., 2004[53]** | I, II, III or IV | diclofenac 100, meloxicam 15 nabumetone 1,000, celecoxib 200 | 10 | 407 | 31.2 | allowed | NR | 24 | 46.9 | 76.9 |
| **Vasey et al., 1987 [57]** | II or III | nabumetone 1,000 naproxen 500 | 5,6,7,8,10 | 318 | 54.4 | allowed | acetaminophen | 24 | 55.0 | NR |
| **Williams et al., 2006[58]** | I, II or III | placebo naproxen 500 | 7,8,10 | 1093 | 59.5 | NR | NR | 12 | 56.2 | 76.4 |
| **Wojtulewski et al., 1996[54]** | I, II or III | meloxicam 7.5 naproxen 750 | 1,3,4,5,6,7,8,10 | 306 | 23.8 | allowed | acetaminophen | 26 | NR | NR |
| **Zhao et al., 2000 [55]** | I, II or III | Placebo celecoxib 100, 200, 400 naproxen 1,000 | 2,9,10 | 688 | 67 | allowed | acetaminophen | 12 | 54.5 | NR |

**Notes.** Outcomes reported: 1: pain; 2: functional disability score; 3: swollen joint count; 4: tender joint count; 5: morning stiffness; 6: grip strength; 7:physician assessment; 8:patient assessment; 9:quality of life scale; 10:adverse events. NR: not reported. Capacity Functional (Class I–Completely able to perform usual activities of daily living; Class II–Able to perform usual self-care and vocational activities but limited in avocational activities; Class III–Able to perform usual self-care activities but limited in vocational and avocational activities; and Class IV–Limited in ability to perform usual self-care, vocational, and avocational activities).

Most of the RCT had attrition bias, since they did not report their results with the intention to analyse reported withdrawals >10% of sample and/or did not discuss the implications of patients lost follow-up, except for five studies [39, 41, 46, 48, 51].

Gibofsky et al. (2007) [44] and Zhao et al. (2000) [55], even though reported adverse events, did not evaluate other important outcomes (such as improvement of pain, swelling and duration of morning stiffness), and thus showed reporting bias. Also, two other clinical trials which described disease diagnostic did not cite the criteria used [56, 57].

**Effect of interventions.**   The doses of drugs were shown as milligrams per day (mg/day).

Nine studies could not be included in the meta-analysis due to absence of standard deviation, standard error or confidence interval data [39, 46–48, 53, 55–58]. All studies were included in the meta-analysis for safety outcome and none of the studies assessed the "disease progression based on radiological imaging of joints" or "satisfaction with current treatment". Subgroup analysis could not be performed due to heterogeneity of the studies.

**Assessed outcomes and network meta-analysis.**   Nine NSAIDs were compared at 13 different dosages and placebo groups. Of the 12 trials, 8 (66.6%) were two-arm studies, whereas 4 (33.3%) were multiple-arm studies (Fig 3). Overall, 4,016 patients were included for "improvement of pain" (Fig 3A); 2,447 patients for "improvement of physical function" (Fig 3B); 4,962 patients for "number of tender/painful joints" (Fig 3C); 4,962 for "number of swollen joints" (Fig 3D), 4,152 for "patient's global assessment" (Fig 3E); and 4,152 for "physician's global assessment" (Fig 3F).

The outcomes of studies that could not be included in the network meta-analysis are described in S2 File.

**Pain.**   This outcome included 13 studies [39–46, 50–52, 54, 59]. Naproxen 1,000 reduced pain compared to placebo (SMD: -10.28, 95% CI: -20.39; -0.17) (evidence of very low quality) (Fig 4, S3 File). Heterogeneity cannot be estimated due to insufficient number of studies. According to the Egger´s test (p = 0.042), the results may be influenced by small-size effects (S6 File).

**Physical function.**   Emery et al. (1999) [59], Furst et al. (2002) [39], Geusens et al. (2004) [43] and Gibofsky et al. (2007) [44] were included in the meta-analysis. Naproxen 1,000 improved physical function compared to placebo (SMD: -0.14, 95% CI: -0.24; -0.05) (evidence of very low quality) (Fig 4, S3 File). Heterogeneity cannot be estimated due to insufficient number of studies. According to the Egger´s test (p = 0.416), the results indicate no obvious publication bias (S6 File).

**Number of tender/painful joints and swollen joints.**   Eight studies were included in the meta-analysis [39, 40, 42–44, 50, 54, 59] while four could not be included [45, 47–49]. The meta-analysis showed significant reduction in number of painful joints for naproxen 1,000 (SMD: -3.54, 95% CI: -5.15; -1.92) (evidence of very low quality) and etoricoxib 90 (SMD: -4.98, 95% CI: -7.13; -2.82) (evidence of very low quality) compared to placebo. Naproxen 750 was better for reducing number of swollen joint than naproxen 1,000 (SMD: -5.21, 95% CI: -9.57; -0.85) (evidence of very low quality), meloxicam 7.5 (SMD: -5.24, 95% CI: -9.11; -1,37) (evidence of low quality), meloxicam 15 (SMD: -6.54, 95% CI: -10.83; -2.25) (evidence of very low quality), meloxicam 22.5 (SMD: -5.34, 95% CI: -9.63; -1.05) (evidence of very low quality), diclofenac 150 (SMD: -6.04, 95% CI: -10.33; -1.75) (evidence of very low quality) and celecoxib 400 (SMD: -5.74, 95% CI: -10.70; -0.78) (evidence of very low quality) (Fig 4, S3 File).

Heterogeneity cannot be estimated due to insufficient number of studies. According to the Egger´s test (p = 0.005), the results may be influenced by small-size effects (S6 File).

**Morning stiffness.**   Twelve studies reported this outcome [41–42, 44–45, 47, 49, 51–52, 54, 56–57, 59], but it was not possible to perform a meta-analysis (S2 File).

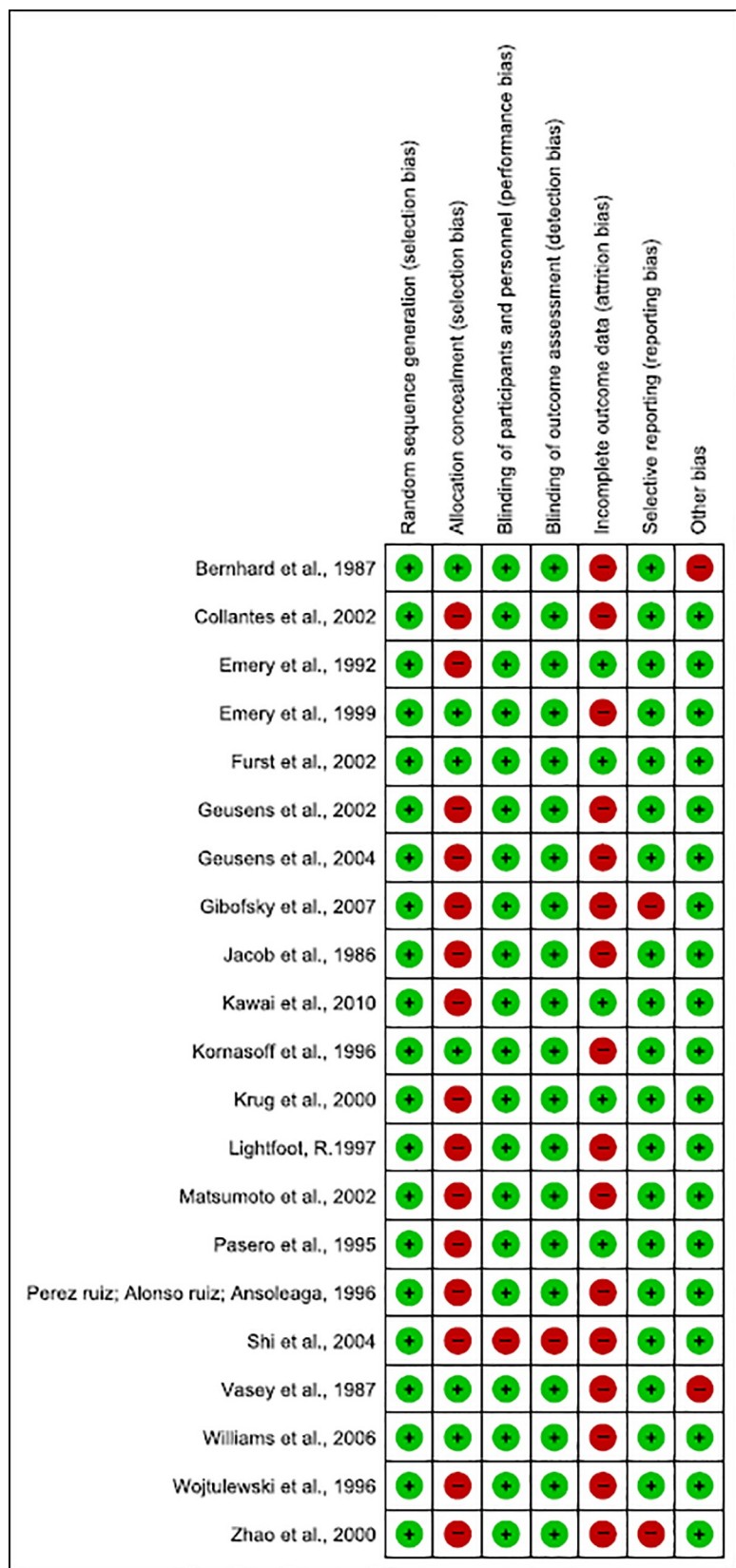

**Fig 2. Risk of bias for studies on non-steroidal anti-inflammatories (n = 21).**

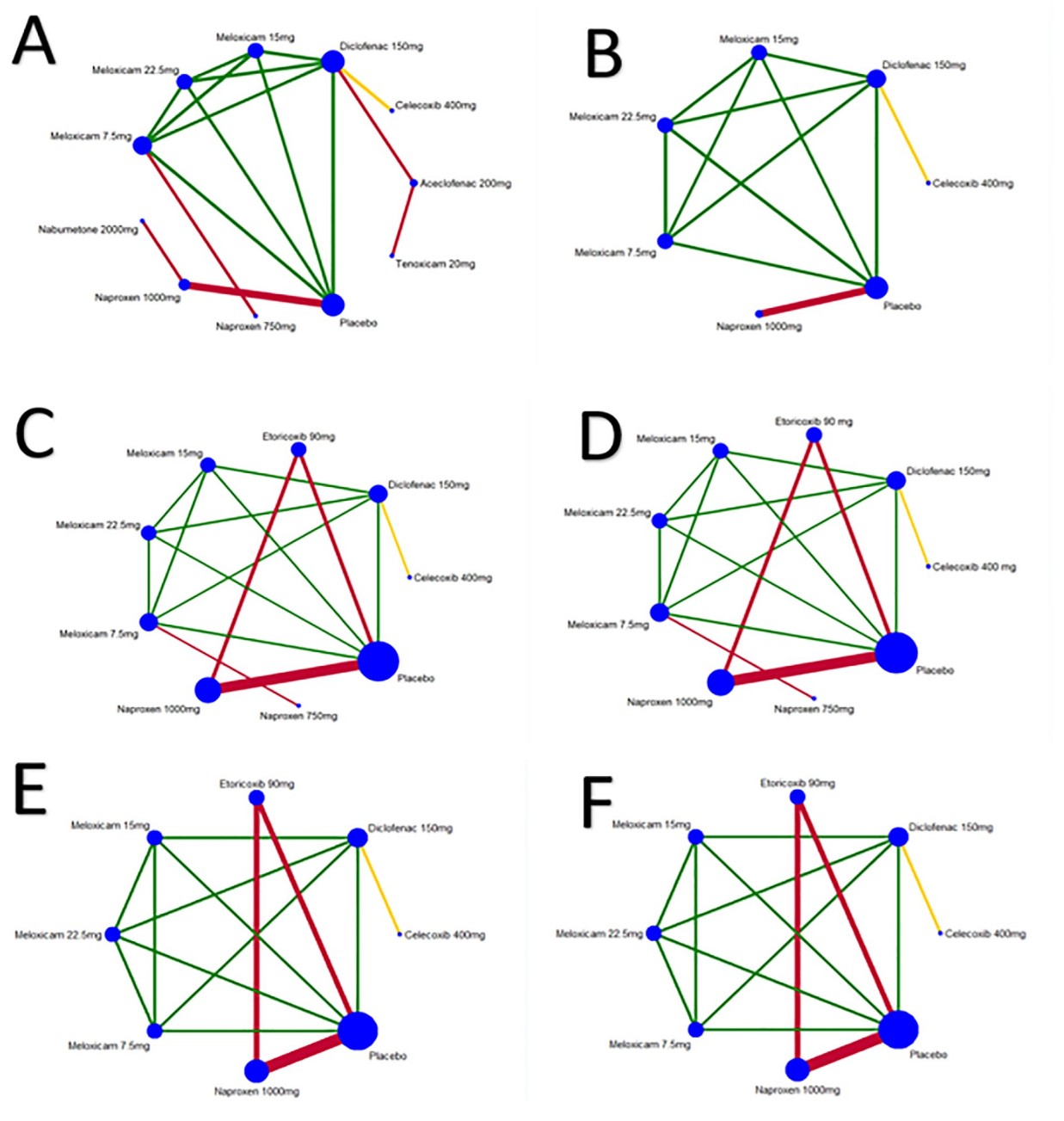

**Fig 3. Evidence structure of eligible comparisons for network meta-analysis: Effectiveness outcomes.** A—Pain (9 studies, 10 non-steroidal anti-inflammatories—NSAIDs, 22 arms, 4,016 patients); B—Physical function (4 studies, 5 NSAIDs, 11 arms, 2,447 patients); C- Number of tender/painful joints (8 studies, 6 NSAIDs, 21 arms, 4,962 patients); D—Number of swollen joints (8 studies, 6 NSAIDs, 21 arms, 4,962 patients); E—Patient's Global Assessment (6 studies, 6 NSAIDs, 17 arms, 4,152 patients); F—Physician's Global Assessment (6 studies, 6 NSAIDs, 17 arms, 4,152 patients).

**Grip strength.** Grip strength changes were assessed in seven studies, but they could not be summarized in a meta-analysis due to improper reporting of data and to use of different drugs [45, 49, 51–52, 54, 56–57] (S2 File).

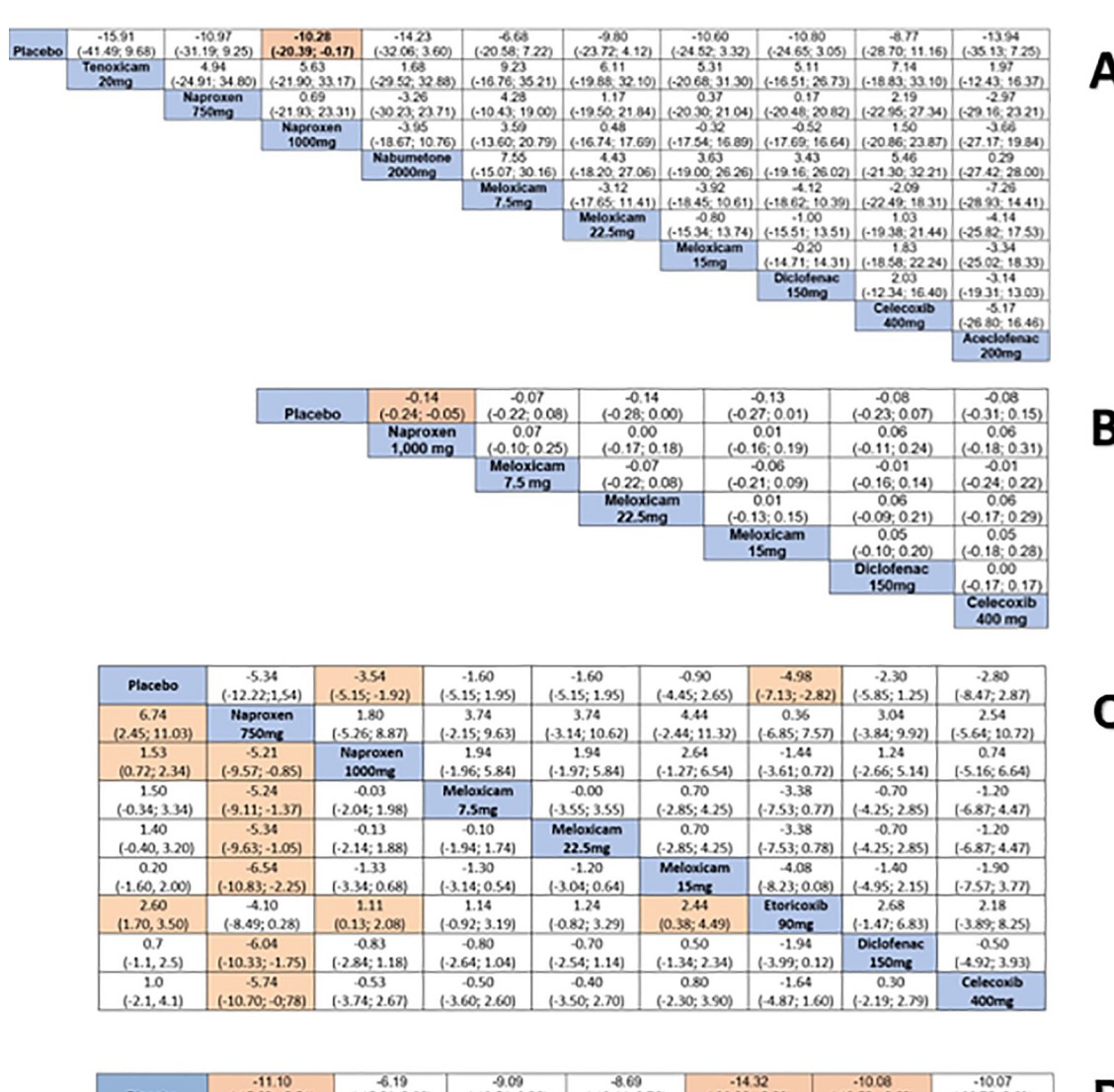

**Fig 4. Comparative effectiveness outcomes between drugs using network meta-analysis.** Comparisons should be read from left to right. Standardized mean difference (SMD) for comparisons are located in the common cell between the column-defining and row-defining treatment. Numbers on highlighted background are statistically significant. A. Pain; B. Physical function; C. Number of tender/painful joints (upper right quarter) and number of swollen joints (lower left quarter); D. Patient's global assessment (upper right quarter) and physician's global assessment (lower left quarter).

**Quality of life.** Two studies [44, 55] investigated this outcome, but meta-analysis was not performed due to improper reporting of data and measurements carried out with different scales (S2 File).

**Patients' and physicians' global assessment.** Six studies which investigated patient's global assessment were included in the meta-analysis [39–40, 43–44, 50, 59]. Other studies were not included due to the manner they were reported (graphs, final percentage improvement or absence of standard deviation values) [42, 45, 47–48, 54, 56, 57–58] (S4 File).

Naproxen 1,000 (SMD: -11.68, 95% CI: -15.68; -6.51) (evidence of very low quality), etoricoxib 90 (SMD: -14.32, 95% CI: -20.26; -8.38) (evidence of low quality) and diclofenac 150 (SMD: -10.08, 95% CI: -19.52; -0.63) (evidence of high quality) were better than placebo for patient's global assessment. Heterogeneity cannot be estimated due to insufficient number of studies. According to the Egger´s test (p = 0.987), the results indicate no obvious publication bias (S6 File).

For physician's global assessment, all drugs assessed were better than placebo, except celecoxib 400. In general, the evidence assessed was of very low to moderate quality; the only exception was for diclofenac 150 *vs* placebo (the evidence was of high quality). Etoricoxib 90 was better than both celecoxib 400 (SMD: -6.28, 95% CI: -12.55; -0.01) (evidence of very low quality) and naproxen 1,000 (SMD: 4.43, 95% CI: 2.01; 6.84) (evidence of low quality) (Fig 4, S3 File). Heterogeneity cannot be estimated due to insufficient number of studies. According to the Egger´s test (p = 0.277), the results indicate no obvious publication bias (S6 File).

**Safety of the interventions.** All studies were included in the meta-analysis. Overall, 10,072 patients reported a number of adverse events for 12 different NSAIDs (56 arms). Of the 21 trials, 10 (41.6%) were two-arm studies and 11 (58.3%) were multiple-arm studies (Fig 5). Although adverse events were reported for most drugs, only etoricoxib 90 was associated with more adverse events compared to placebo (RR: 4.43, 95% CI: 1.22; 16.08) (evidence of low quality) (Fig 6). Heterogeneity cannot be estimated due to insufficient number of studies. According to the Egger´s test (p = 0.718), the results indicate no obvious publication bias (S6 File).

In most of the studies, gastrointestinal adverse events were the most commonly reported by patients using NSAIDs. Abdominal pain, diarrhea, dyspepsia and nausea were the most frequent events reported in 18 studies [39–45, 47–50, 52–54, 56–59]. NSAIDs responsible for the highest incidence of these events were diclofenac [39, 53, 59] and naproxen [40–44, 48, 50, 58] at any dose.

Hypertension [40, 42, 44] and headache [40, 44, 49, 57–59] were commonly reported adverse events. No study reported serious adverse events leading to death or hospitalization. Early discontinuation due to treatment failure or to adverse events did not differ statistically between the groups and was not associated with any specific NSAID.

**Ranking of treatments and outcomes (Table 2).** Values of SUCRA provided the hierarchy of 24 treatments on the outcomes assessed based on absolute rank probabilities. Tenoxicam 20, nabumetone 2,000 and aceclofenac 200 were most effective at reducing pain (4.3%, 4.6% and 4.8%, respectively).

Best results for improvement of physical function were observed for naproxen 1,000 (2.6%), meloxicam 22.5 (2.7%) and meloxicam 15 (3.0%).

Regarding number of tender/painful joints and swollen joints, etoricoxib 90 (2.1%) and naproxen 750 (1.1%) had the highest improvement rates, respectively.

As for patient's global assessment and physician's global assessment, etoricoxib 90 was considered the best intervention for both variables (2.0% and 1.2%, respectively). Celecoxib 200, placebo and nabumetone 2,000 were associated with a smaller number of adverse events and had the best safety profile (6.0%, 7.8% and 7.9%, respectively).

## Corticoids for rheumatoid arthritis

**Description of studies.** Five trials (1,544 patients) were included. No study reported the functional class of the disease. These studies investigated the drugs prednisone 5, 7.5, 10 and

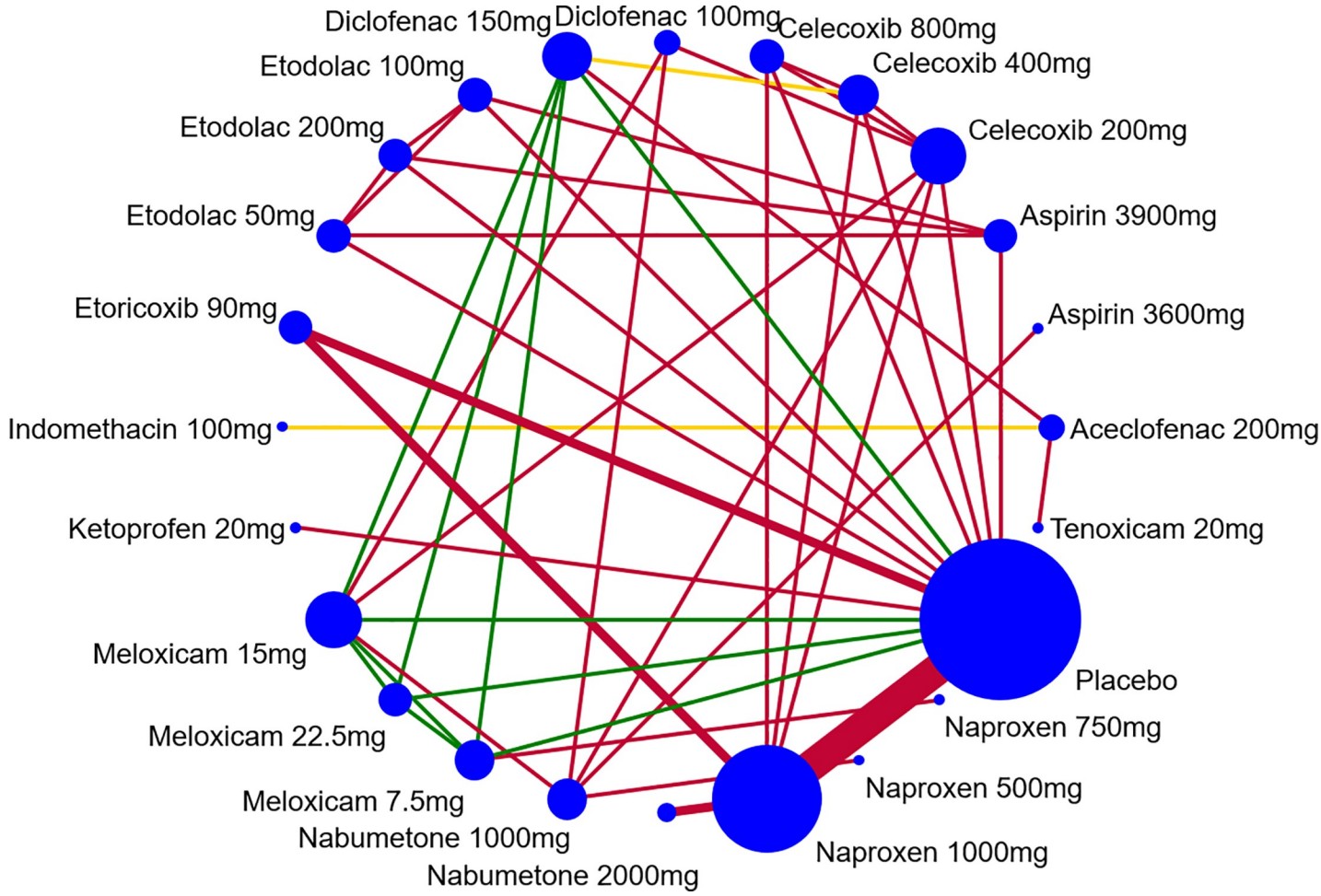

**Fig 5. Evidence structure of eligible comparisons for network meta-analysis: Adverse events.** Lines connect the interventions that have been studied in head-to-head (direct) comparisons in the eligible randomized controlled trials. The width of the lines represents the cumulative number of randomized controlled trials for each pairwise comparison and the size of every node is proportional to the number of randomized participants (sample size).

15 and prednisolone 7.5, administered orally. Patient follow-up ranged from 12 to 104 weeks. No studies reported the use of rescue medication and four studies described concomitant use of DMARD therapy. The mean age of the participants ranged from 39.9 to 58 years (Table 3).

**Risk of bias (Fig 7).** All of the assessed trials considered eligible were at high risk of bias, except for Choy et al. (2008) [11] which was at minimum risk of bias. Allocation concealment was insufficiently described in four studies [12, 60–62]. One study did not describe the blinding of patients and healthcare professionals and failed to report whether there were patients lost at follow-up [60].

Bakker et al. (2012) [12], Buttegereit et al. (2013) [60], Ding et al. (2012) [61] and Hafstrom et al. (2014) [62] did not describe other important outcomes, such as reduction of pain, swelling and duration of morning stiffness or the main adverse events reported by patients, leading to reporting bias. Two clinical trials did not cite diagnostic criteria used [12, 62].

**Effect of interventions.** One placebo-controlled RCT of 2 years investigated whether adding prednisone 10 to the therapy increased effectiveness of methotrexate (dose was increased by 5 mg/week until remission) for the early rheumatoid arthritis. Erosive joint damage was significantly lower for the methotrexate and prednisone group (p<0.022) compared to the

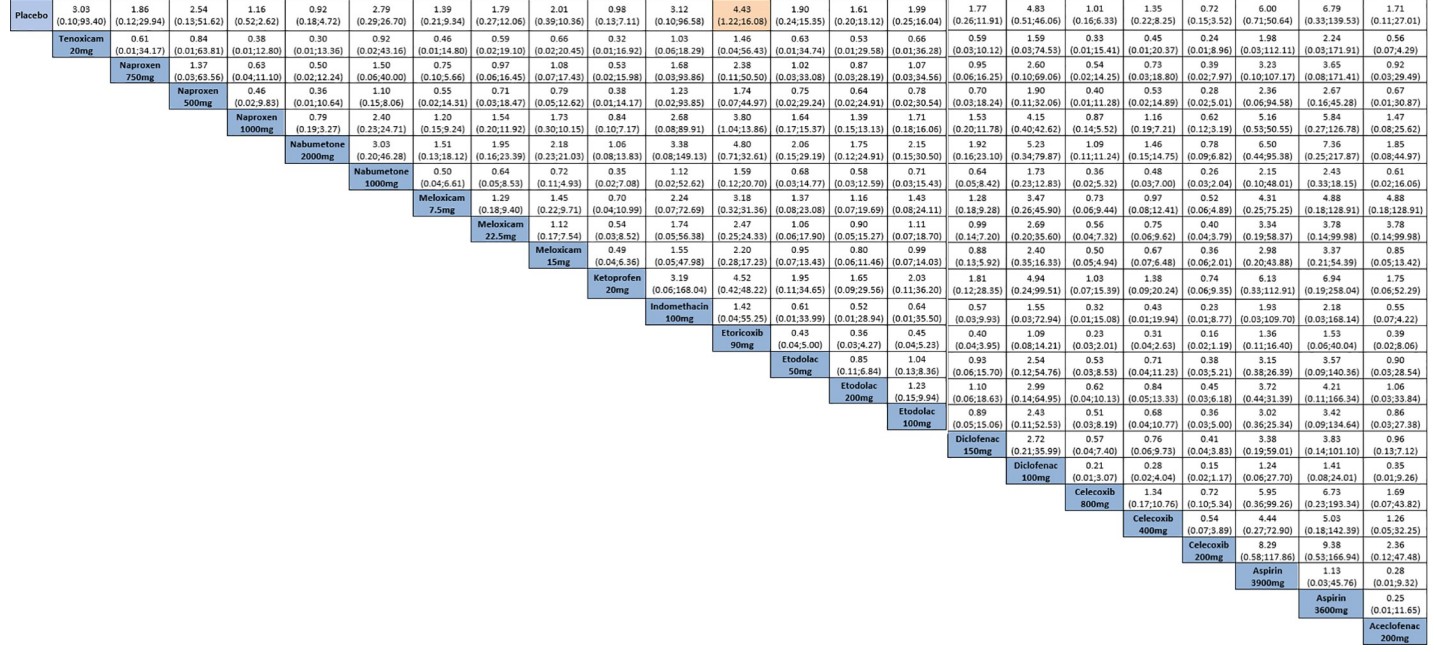

**Fig 6. Comparative adverse events between drugs using network meta-analysis.** Comparisons should be read from left to right. Odds ratio for comparisons are located in the common cell between the column-defining and row-defining treatment. Numbers on highlighted background are statistically significant.

control group (methotrexate and placebo). Adverse events were similar for both groups with the most frequent gastrointestinal-related events being nausea, diarrhea and stomachache, whereas central nervous system events included headache, dizziness and blurred vision. The number of SAE also did not differ between groups [12].

In another RCT, 350 patients received prednisone 5 or placebo. The authors showed that prednisone increased the proportion of patients achieving low disease activity (p = 0.0109) and that the incidence of adverse events was similar for both groups (7.8% vs 8.4%). In both groups, the most frequent adverse events were related to worsening of the disease and to arthralgia and occurred statistically more frequently in the placebo group (p = 0.0141). The most reported events after arthralgia were nasopharyngitis, headache, hypertension and diarrhea [60].

Choy et al. (2008) [11] randomized patients into four groups: methotrexate (starting at 7.5 mg/week, increasing incrementally up to target dose of 15 mg/week); prednisolone (starting at 60 mg/day, reduced to 7.5 mg at the 6th week, 7.5 mg daily from the 6th to the 28th week, stopped at the 34th week); cyclosporin (started 3 months after methotrexate treatment at initial dose of 100 mg/day, increased gradually up to target dose of 3 mg/kg daily); and triple therapy (methotrexate, prednisolone and cyclosporin). Erosions were reduced due to the use of prednisolone (p = 0.03) and cyclosporin (p = 0.01). There was improvement in quality of life for all treatments. Great number of patients on triple therapy left the study due to adverse events.

One RCT assessed outcome safety in 266 patients randomized in three groups: placebo, prednisone 7.5 and prednisone 15. All groups used concomitantly leflunomide 20 and methotrexate 10. The combination therapy with prednisone 7.5, leflunomide 20 and methotrexate 10 showed that the incidence of adverse events (skin rash, liver dysfunction and oral ulcers) decreased. The authors concluded that this therapy could be a useful option for initial treatment of early rheumatoid arthritis [61].

**Table 2. Ranking of treatments for the outcomes assessed.** Bold numbers on highlighted background are first in ranking. For the safety outcome, the highlight represents the safety. For effectiveness outcomes, the highlight represents the most effective.

| Interventions | Safety | | Improvement of pain | | Improvement of physical function | | Number of tender/ painful joints | | Number of swollen joints | | Patient's Global Assessment | | Physician's Global Assessment | |
|---|---|---|---|---|---|---|---|---|---|---|---|---|---|---|
| | SUCRA | Mean Rank (SE) | SUCRA | Mean Rank (SE) | SUCRA | Mean Rank (SE) | SUCRA | Mean Rank (SE) | SUCRA | Mean Rank (SE) | SUCRA | Mean Rank (SE) | SUCRA | Mean Rank (SE) |
| Celecoxib 200 mg | 78.1 | **6.0 (0.3)** | - | - | - | - | - | - | - | - | - | - | - | - |
| Placebo | 70.4 | 7.8 (0.3) | 10.7 | 9.9 (0.2) | 10.7 | 6.4 (0.1) | 12.9 | 8.0 (0.2) | 13.0 | 8.0 (0.2) | 4.0 | 7.7 (0.1) | 2.3 | 7.8 (0.1) |
| Nabumetone 2.0g | 69.9 | 7.9 (0.3) | 63.9 | 4.6 (0.2) | - | - | - | - | - | - | - | - | - | - |
| Ketoprofen 20 mg | 66.9 | 8.6 (0.4) | - | - | - | - | - | - | - | - | - | - | - | - |
| Celecoxib 800 mg | 66.1 | 8.8 (0.3) | - | - | - | - | - | - | - | - | - | - | - | - |
| Naproxen 1.0g | 64.3 | 9.2 (0.3) | - | - | 73.2 | **2.6 (0.1)** | 65.9 | 3.7 (0.1) | 54.8 | 4.6 (0.1) | 63.9 | 3.5 (0.1) | 52.8 | 4.3 (0.1) |
| Meloxicam 7.5 mg | 60.1 | 10.2 (0.4) | 35.9 | 7.4 (0.2) | 37.0 | 4.8 (0.1) | 36.2 | 6.1 (0.2) | 55.2 | 4.6 (0.1) | 31.0 | 5.8 (0.1) | 44.3 | 4.9 (0.1) |
| Celecoxib 400 mg | 59.3 | 10.4 (0.4) | 45.2 | 6.5 (0.2) | 45.6 | 4.3 (0.1) | 55.1 | 4.6 (0.2) | 42.1 | 5.6 (0.2) | 57.7 | 4.0 (0.1) | 30.0 | 5.9 (0.1) |
| Etodolac 200 mg | 52.9 | 11.8 (0.4) | - | - | - | - | - | - | - | - | - | - | - | - |
| Aceclofenac 200g | 52.3 | 12.0 (0.4) | 62.4 | 4.8 (0.2) | - | - | - | - | - | - | - | - | - | - |
| Naproxen 750 mg | 51.2 | 12.2 (0.4) | 55.2 | 5.5 (0.2) | - | - | 79.6 | 2.6 (0.2) | 99.2 | **1.1 (0.1)** | - | - | - | - |
| Diclofenac 150 mg | 51.0 | 12.3 (0.4) | 55.0 | 5.5 (0.2) | 43.9 | 4.4 (0.1) | 50.0 | 5.0 (0.1) | 32.4 | 6.4 (0.1) | 58.3 | 3.9 (0.1) | 57.9 | 3.9 (0.1) |
| Meloxicam 22.5 mg | 50.4 | 12.4 (0.4) | 49.5 | 6.0 (0.2) | 72.2 | 2.7 (0.1) | 37.8 | 6.0 (0.2) | 52.8 | 4.8 (0.1) | 52.4 | 4.3 (0.1) | 58.7 | 3.9 (0.1) |
| Etodolac 100 mg | 49.0 | 12.7 (0.4) | - | - | - | - | - | - | - | - | - | - | - | - |
| Meloxicam 15 mg | 49.3 | 12.7 (0.4) | 53.9 | 5.6 (0.2) | 67.4 | 3.0 (0.1) | 26.6 | 6.9 (0.2) | 18.7 | 7.5 (0.1) | 47.7 | 4.7 (0.1) | 57.4 | 4.0 (0.1) |
| Etodolac 50 mg | 48.7 | 12.8 (0.4) | - | - | - | - | - | - | - | - | - | - | - | - |
| Naproxen 500 mg | 43.8 | 13.9 (0.4) | - | - | - | - | - | - | - | - | - | - | - | - |
| Nabumetone 1.0g | 40.8 | 14.6 (0.4) | 51.4 | 5.9 (0.2) | - | - | - | - | - | - | - | - | - | - |
| Indomethacin 100mg | 38.6 | 15.1 (0.4) | - | - | - | - | - | - | - | - | - | - | - | - |
| Tenoxicam 20 mg | 38.9 | 15.1 (0.4) | 67.1 | **4.3 (0.2)** | - | - | - | - | - | - | - | - | - | - |
| Diclofenac 100 mg | 26.8 | 17.8 (0.4) | - | - | - | - | - | - | - | - | - | - | - | - |
| Etoricoxib 90 mg | 25.8 | 18.1 (0.4) | - | - | - | - | 86.1 | **2.1 (0.1)** | 81.9 | 2.4 | 85.0 | **2.0 (0.1)** | 96.6 | **1.2 (0.1)** |
| Aspirin 3.600 mg | 24.1 | 18.5 (0.4) | - | - | - | - | - | - | - | - | - | - | - | - |
| Aspirin 3.900 mg | 21.5 | 19.1 (0.4) | - | - | - | - | - | - | - | - | - | - | - | - |

Notes. SUCRA: surface under the cumulative ranking curve. Mean Rank: average ranking of treatments according to their relative effectiveness. The first ranked treatment is most likely to be the most effective treatment regarding a particular outcome compared to other treatments in the network. Numbers on highlighted background are statistically significant. SE: standard error.

**Table 3. Characteristics of the studies on corticoids anti-inflammatories for rheumatoid arthritis (n = 5).**

| Study | Functional capacity | Interventions | Outcomes reported | Sample size (N) | Lost follow-up (%) | DMARD Use | Rescue medication | Duration (weeks) | Mean age (years) | Woman (%) |
|---|---|---|---|---|---|---|---|---|---|---|
| Bakker et al., 2012[12] | NR | methotrexate* and prednisone 10, methotrexate* and placebo | 1,2,5 | 236 | 27.9 | allowed | NR | 52 | 53.5 | 60.1 |
| Buttgereit et al., 2013 [60] | NR | prednisone 5, placebo | 2,5 | 350 | 7.7 | allowed | NR | 12 | 57.3 | 84 |
| Choy et al., 2008[11] | NR | methotrexate alone** methotrexate** and ciclosporin*** methotrexate** and prednisolone methotrexate**, ciclosporin*** and prednisolone$ | 1,2,3,4,5 | 467 | 18.8 | allowed | NR | 52 | 54 | 69.5 |
| Ding et al., 2012[61] | NR | prednisone 7.5, prednisone 15 placebo$$ | 5 | 266 | 5.6 | not allowed | NR | 12 | 43 | 85.3 |
| Hafstrom et al., 2014 [62] | NR | prednisolone 7.5 placebo | 1 | 225 | 46.2 | allowed | NR | 104 | 54.5 | 64 |

Notes.

*dose increased by 5 mg/week until remission

**starting at 7.5 mg/week, increasing incrementally up to target dose of 15 mg/week

***ciclosporin started 3 months after methotrexate (initial dose 100 mg/day, increased gradually up to target dose of 3 mg/kg daily)

$60 mg/day initially, reduced to 7.5 mg daily from 6 to 28 weeks, stopped by week 34

$$all groups received leflunomide 20 mg/day and methotrexate 10 mg/day; Outcomes reported: 1. Progression of the disease assessed by radiological imaging of joints; 2. Disease activity; 3. Function; 4. Quality of life; 5. Adverse events. NR: not reported. Capacity Functional (Class I–Completely able to perform usual activities of daily living; Class II–Able to perform usual self-care and vocational activities but limited in avocational activities; Class III–Able to perform usual self-care activities but limited in vocational and avocational activities; and Class IV–Limited in ability to perform usual self-care, vocational, and avocational activities).

One RCT assessed the predictors of radiographic progression in 225 patients treated with or without prednisolone. The study showed that the frequency of patients with radiographic progression after 2 years was smaller (26%) for the prednisolone group 7.5 in comparison to the placebo group (39%) (p = 0.033) [62].

# Discussion

## Summary of evidence and comparison of findings with previous studies

This systematic review assessed the available evidence on effectiveness and safety of NSAIDs and corticoids for the treatment of rheumatoid arthritis. RCT assessed the use of NSAIDs, mainly via oral administration. The main methodological flaws were absence of allocation concealment and high dropout rates; it was also often unclear how analyses were performed or how the studies dealt with missing data.

In general, it was observed that naproxen 1,000 improved physical function and reduced overall pain and number of painful and swollen joints, providing benefits according to the patient's and physician's global assessment compared to placebo. Also, naproxen 750 was better than most of the NSAIDs at reducing the number of swollen joints (including naproxen 1,000), except in comparison to etoricoxib 90. However, the quality of evidence was very low overall. It can be observed that naproxen did not exhibit a dose-dependent behaviour as the 750 dose was more effective than the 1,000 dose for the number of swollen joints outcome.

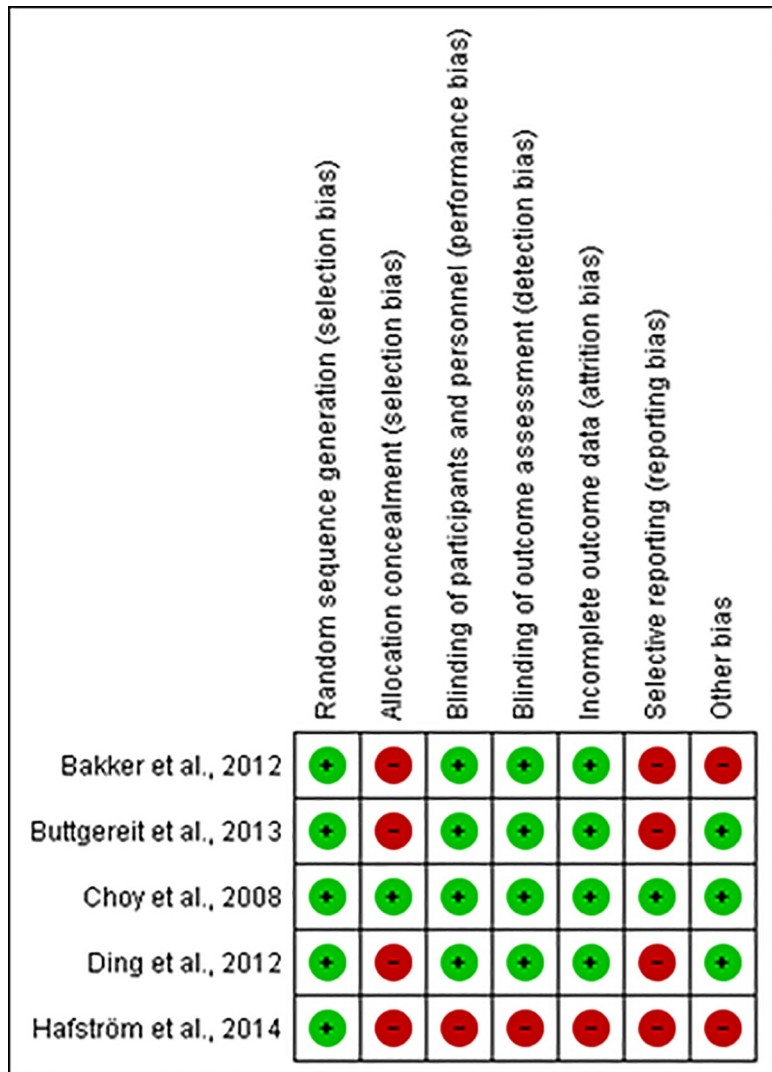

**Fig 7. Risk of bias for studies on corticoids (n = 5).**

Etoricoxib 90 was also reported to reduce the number of painful and swollen joints and provided better results according to the patient's global assessment compared to placebo. In addition, it was better than both celecoxib 400 and naproxen 1,000, according to physician's global assessment, but the quality of evidence was very low.

No study using NSAIDs reported "progression of disease as assessed by radiological imaging of joints". Also, it was not possible to perform meta-analysis for the outcomes morning stiffness, average grip strength and improvement of quality of life.

In another review [63], celecoxib 200 showed significant pain reduction compared to placebo (11% absolute improvement; 95% CI 8% to 14%), which was not evidenced in our review. However, this study also related low quality of the evidence of your findings.

Etoricoxib 90 improved overall outcomes when compared to placebo, but did not have a good safety profile according to the ranking hierarchy of 24 treatments. Celecoxib 200, nabumetone 2,000 and placebo were associated with a smaller number of adverse events. The most

common side effects reported by patients using NSAIDs were gastrointestinal events and no SAE were documented.

Findings from safety data of another systematic review showed that piroxicam or etodolac did not lead to clinically significant adverse events and that mild adverse events were evidenced by the use of celecoxib or etoricoxib such as nausea, vomiting and headaches. However, the quality of the evidence did not allow confirmation as to which NSAID is safest [6].

Evidence of moderate to low quality reported fewer withdrawals among patients which had received celecoxib compared to those receiving placebo. Celecoxib 200 was associated with a smaller number of adverse events, but it was observed that patients developed more ulcers and severe short-term symptoms [63].

Prednisolone 7.5 and prednisone 5.0, 7.5 and 10 showed benefits in reduction of joint erosion [11, 12, 62] and disease activity [12, 60]. Due to different interventions employed in these clinical trials, it was not possible to perform meta-analysis. Besides that, issues regarding risk of bias decreased reliability of such findings.

Regarding safety of corticoids, gastrointestinal adverse events were the issues most often reported by patients using these drugs. Corticoids are usually used in association with methotrexate, and thus attention should be focused on the adverse effects of combined therapy.

The effects of prednisone or similar corticoid preparations administered alongside standard therapy (doses ranging from 270mg to 5,800mg, over the first year) reduced progression of joint erosion in rheumatoid arthritis, according to a systematic review published in 2007 [22]. Only one clinical trial included in this review [11] was also selected for our study due to differences in eligibility criteria. In our findings, two studies used prednisone at doses of 5–10 demonstrating benefit of this drug for this outcome [61, 62].

Systematic review showed that prednisolone administrated orally at low doses (maximum of 15 mg/day) was more effective than placebo and NSAIDs at decreasing joint tenderness and pain, but the quality of the evidence was not assessed by the authors [23]. According to older evidences, after six months, the effects of prednisolone (maximum 15 mg/day) were significantly better compared to placebo for number of painful joints, number of swollen joints, pain and physical function [21]. However, small sample size, high risk of bias and the fact that quality of evidence was not assessed compromise such findings.

## Strengths and limitations of this study

The methodology employed in this review includes explicit eligibility criteria, comprehensive and extensive database searches and independent and paired evaluation to select studies. Meticulous search and selection processes were carried out, and we are confident that trials meeting the inclusion criteria were included in the review. Robust statistical techniques were used to assess risk of bias of the included studies.

The decision to provide network analysis is relevant as it provides information in situations where primary evidence is scarce or nonexistent and allows for more accurate estimates of effects [64]. Even though a broad search strategy was carried out and that we did not exclude any studies due to language barriers or to date of publication, some bias cannot be considered inexistent.

For some studies that did not provide variance measures necessary for meta-analysis, we estimated missing data with approximate values derived graphically from the studies themselves. This could have created some bias, but the overall impact on the estimation of statistically significant differences between groups is probably small.

Some studies did not record the concomitant use of other analgesic agents and/or the dose used, which can mislead outcomes for pain measurement, for example.

It's important to highlight that anti-inflammatories drugs are still used as adjuncts to control symptoms of disease. However, the impact of use mainly of NSAIDs has decreased during the recent years, since more potent DMARDs allow for better disease control as well as glucocorticoids are mainly used for bridging while the effects of DMARDs take place.

The quality of the primary studies included in this review was a limiting factor for proper analysis to be carried out. Besides that, the diversity of drugs and doses used and absence of report of some outcomes could have decreased the quality of our findings.

### Implications for clinical practice and research

NSAIDs and corticoids are used as adjuncts to treat sympton of rheumatoid arthritis. Therefore, these anti-inflammatory drugs should not be used for long-term. According to our findings, even though etoricoxib 90 and naproxen 1,000 appear to be effective and celecoxib 200 seems to be the safest option, the low quality of the evidence suggests that futures RCT can demonstrate different results.

Prednisone and prednisolone seem to be effective and generate only mild adverse events in patients with rheumatoid arthritis, when used at lower doses. However, due to the high risk of bias of RCTs and not being possible to perform the meta-analysis of the studies involving these drugs, the evidence of such findings could not be confirmed.

We observed that the clinical trials had methodological limitations, differences regarding the drugs studied and their doses, concurrent use with other drugs and differents outcomes measured contributed to limiting the conclusions on our findings. Future trials should consider these limitations and therefore obtain long-term data. Also, adequate follow-up and larger sample sizes are required for future research.

### Conclusion

Naproxen, prednisolone and prednisone were considered the most effective drugs and celecoxib showed fewer adverse events. However, the low quality of the evidence observed for the outcomes of NSAIDs, the absence of meta-analyses to assess the outcomes of corticoids and the risk of bias observed in clinical trials indicate that future RCT can confirm such findings.

### Supporting information

**S1 Checklist.**
(DOCX)

**S1 File. Search strategies for different databases.**
(DOCX)

**S2 File. Main outcomes found in the articles not included in the meta-analysis.**
(DOCX)

**S3 File. GRADE for effectiveness and safety outcomes.**
(DOCX)

**S4 File. Reason for exclusion of studies.**
(DOCX)

**S5 File. Studies included in the review.**
(DOCX)

**S6 File. Heterogeneity of studies according to analysed outcomes.**
(DOCX)

## Author Contributions

**Conceptualization:** Mariana Del Grossi Paglia, Silvio Barberato-Filho, Flavia Casale Abe, Cristiane de Cássia Bergamaschi.

**Data curation:** Mariana Del Grossi Paglia, Marcus Tolentino Silva, Lauren Giustti Mazzei, Cristiane de Cássia Bergamaschi.

**Formal analysis:** Marcus Tolentino Silva.

**Investigation:** Mariana Del Grossi Paglia, Silvio Barberato-Filho, Lauren Giustti Mazzei, Flavia Casale Abe, Cristiane de Cássia Bergamaschi.

**Methodology:** Mariana Del Grossi Paglia, Luciane Cruz Lopes, Lauren Giustti Mazzei, Flavia Casale Abe, Cristiane de Cássia Bergamaschi.

**Project administration:** Luciane Cruz Lopes, Cristiane de Cássia Bergamaschi.

**Resources:** Mariana Del Grossi Paglia, Silvio Barberato-Filho.

**Supervision:** Luciane Cruz Lopes, Cristiane de Cássia Bergamaschi.

**Validation:** Mariana Del Grossi Paglia, Luciane Cruz Lopes, Cristiane de Cássia Bergamaschi.

**Visualization:** Mariana Del Grossi Paglia, Luciane Cruz Lopes, Cristiane de Cássia Bergamaschi.

**Writing – original draft:** Mariana Del Grossi Paglia, Cristiane de Cássia Bergamaschi.

**Writing – review & editing:** Mariana Del Grossi Paglia, Marcus Tolentino Silva, Luciane Cruz Lopes, Silvio Barberato-Filho, Lauren Giustti Mazzei, Flavia Casale Abe, Cristiane de Cássia Bergamaschi.

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
