## [Decision Letter · Decision Letter 0]

12 Nov 2020

PONE-D-20-24917

Use of corticoids and non-steroidal anti-inflammatories in the treatment of rheumatoid arthritis: systematic review and network meta-analysis

PLOS ONE

Dear Dr. Paglia,

Thank you for submitting your manuscript to PLOS ONE. After careful consideration, we feel that it has merit but does not fully meet PLOS ONE’s publication criteria as it currently stands. Therefore, we invite you to submit a revised version of the manuscript that addresses the points raised during the review process.

We look forward to receiving your revised manuscript.

Kind regards,

Ivan D. Florez

Academic Editor

PLOS ONE

Journal Requirements:

2. Thank you for stating the following above the Acknowledgments Section of your manuscript:

'Funding This project is funded by governmental Program Graduate Education Institutions— PROSUP—CAPES/UNISO.'

'The authors received no specific funding for this work.'

4. Please include your tables as part of your main manuscript and remove the individual files. Please note that supplementary tables should remain as separate "supporting information" files.

5. Please include captions for your Supporting Information files at the end of your manuscript, and update any in-text citations to match accordingly. Please see our Supporting Information guidelines for more information: http://journals.plos.org/plosone/s/supporting-information

Additional Editor Comments:

Your manuscript has been reviewed by two experts in the field, and they have found some points that need to be addressed before this manuscript is considered for publication. Additionally, I have performed an Editorial Review. Please go through the reviewers' and Editor's comments, consider addressing these points, and prepare a revised version.

- Authors do not mention how they assessed and handled the transitivity. This is a major issue as transitivity is one of the core assumptions of NMAs. If not met, the validity of the results is threatened. If this assessment is not present, this should be added as a limitation of the paper.

-Authors state they applied GRADE NMA approach to assess the quality. However, I see several issues in this application.

1) as stated above, how they applied intransitivity, is not clear, is not even mentioned in the paper, and this is one of the GRADE criteria.

2) Puhan's paper, although described the approach, it is not the only guidance, at least 2 more papers have been published by the GRADE group: Brignardello-Petersen et al 2017 and 2018 that should be applied, and they were not

3) In appendix C, there are several issues, such as: why there are not Superscripts in the NMA GRADE assessment? The superscripts used in the indirect assessments show the reasons for downgrading the quality/certainty. in the NMA assessment, according to the GRADE approach, there are at least 2 criteria to be applied (Puhan and Brignardello-Petersen's paper) and they seem to be neglected as none of the NMA has a superscript to describe this.

4) again in appendix C, lets take an example with comparison "Placebo v naproxen 1,000 mg". Authors state that in the indirect evidence, there was "inconsistency". A huge problem here, is that inconsistency is evaluated as part of the direct assessment. So this might mean these criteria seems to be the reasons for rating down the direct evidence that informs the indirect estimate. Thus, there is no information to evidence that the Puhan et al. approach was applied, as the criteria for doing that with indirect evidence, is the intransitivity criterion, which, again, was not applied

5) As a result of the flaws identified in the GRADE assessment, there is no quality assessment of the NMA estimates, which are the most important ones in NMA. It seems authors only applied the GRADE direct criteria (traditional GRADE assessment), but the indirect and NMA estimates were neglected. Thus, the authors should not describe that they applied the Puhan's approach.

In summary to resolve points, 1-5 described above, you should fully apply the GRADE NMA approach as described by Puhan as you stated (complemented by Brignardello-Petersen's papers 2017 and 2018 (GRADE PAPERS from J of Clin Epidemiol). IF this is not possoble as this time, this should be highlighted as a limitation.

NOTE: two new GRADE approaches by Briganrdello_Petersen et al, were published very recently in the BMJ 2020 (PMID: 33177059 and PMID: 33172877), are not mandatory to apply as they were not available when you performed your analysis.

-Table 2 shows the SUCRA Values and the mean rank. The Mean rank was neither explained in methods, nor in the table. I think the mean rank might be misleading, SUCRA Values (and an explanation, a brief one, in a footnote of the table might be enough). If you want to show the mean, then a variability measure (e.g., SD, 95%CI) should be added.

Reviewers' comments:

Reviewer's Responses to Questions

**Comments to the Author**

1. Is the manuscript technically sound, and do the data support the conclusions?

Reviewer #1: Partly

Reviewer #2: Partly

2. Has the statistical analysis been performed appropriately and rigorously? 

Reviewer #1: Yes

Reviewer #2: I Don't Know

3. Have the authors made all data underlying the findings in their manuscript fully available?

Reviewer #1: Yes

Reviewer #2: Yes

4. Is the manuscript presented in an intelligible fashion and written in standard English?

Reviewer #1: Yes

Reviewer #2: Yes

5. Review Comments to the Author

Reviewer #1: The authors provide a systematic review of NSAIDs and glucocorticoids (GC) in the treatment of rheumatoid arthritis (RA). For the NSAIDs a network meta-analysis (NMA) is done.

To establish indirect comparisons in NMA, similarity, homogeneity and consistency must be assessed. In this paper, there is only scarce information about the similarity of the mentioned NSAID studies especially concerning disease activity (as measured by the number of swollen joints, ESR etc.), functional capacity, concomitant DMARD treatment.

Although NSAIDs are still used in the treatment of RA, their impact has decreased during the recent years as more potent DMARDs allow for better disease control. Therefore NSAIDs are only used as an adjunct. This might be stressed in the discussion part.

Glucocorticoids are mainly used for bridging and their long-term use should be avoided. This should also be mentioned in the discussion part to underline the heterogeneity of the selected GC studies.

Reviewer #2: This is a valuable piece of work and admirable that this is done without having any specific funding. I also checked that this is a unique research. There are about 60 other published NMA about RA but none of them have targeted all these medications in one place. I am providing some critical comments that I hope could help the authors and support the editor's decision.

A. Search Step

1. I noticed that the authors have not used any of the established RCT search filters that are freely available and are being used as gold standard to retrieve the RCT studies from most of the databases. This seems to be a major issue as the performance of these filters shows that they are the best way of searching RCTs. All these filters are freely available as an appendix for the Search chapter of Cochrane handbook.

2. We usually used Peer-Review of Electronic Search Strategies checklist (PRESS) to peer-review the search at protocol stage. The authors have not followed the items from the PRESS checklist. There are no Controlled Vocabularies used in reported search strategies. In addition, the name of medications/drugs have not been searched at all and the search only contains the class of drugs.

3. The authors report that they have searched CENTRAL however they have reported 'Cochrane' in appendix. If by Cochrane they mean Cochrane Library it is different from CENTRAL.

4. The authors report that they have searched Web of Science however they did not mention the names of the databases that have been searched via Web of Science:

Science Citation Index Expanded (SCI-EXPANDED) --1900-present

Social Sciences Citation Index (SSCI) --1956-present

Arts & Humanities Citation Index (A&HCI) --1975-present

Conference Proceedings Citation Index- Science (CPCI-S) --1990-present

Conference Proceedings Citation Index- Social Science & Humanities (CPCI-SSH) --1990-present

Emerging Sources Citation Index (ESCI) --2015-present

5. The authors report that they have searched CINAHL via Ovid SP while since 2006, CINAHL is only available via EBSCOhost.

6. There are discrepancies among PROSPERO protocol, published protocol in Medicine (Baltimore), and the final review submitted to PLoS One in terms of databases and search strategies:

6.1 The PROSPERO records mentions Scopus as one of the databases to be searched but Scopus is missing from the published protocol and submitted review for PLoS One.

6.2 Furthermore, the search strategy reported in published protocol in Medicine (Baltimore) is for PubMed. Not only is PubMed not among the searched resources but also the search strategy is different from what reported in the final review for MEDLINE or other databases. Although both search strategies are questionable for not including RCT filters and not following PRESS checklist, one major difference is that the search strategies in the final review do not contain any drug names while the protocol does.

6.3 In published protocol, the authors write in the text that they report search strategy for MEDLINE (via Ovid) in Table 1 and then they report search strategy for MEDLINE (via PubMed). In terms of syntax, the reported search strategy can be run in both interfaces – regardless of its issues – but the discrepancies shows that the search has not been peer-reviewed.

B. Screening

1. In the PRISMA flow diagram, there are 7615 results after removing the duplicates however the authors only screened 192 records for title and abstract. It is unclear what happens for the remaining 7423 records.

2. Again in full-text screening, the authors screen 86 full-text and exclude 41 with reason that leaves 45 to include. However, the authors include only 26 not 45. So it is unclear what happens to the remaining 19.

I only checked the parts of the manuscript that was related to my area and I leave the rest to the relevant reviewers/editors to decide.

Again I want to state that I am impressed with the work done by a determined team for a non-funded project such as this one and I hope the authors find the comments helpful for this review and their future reviews.

6. PLOS authors have the option to publish the peer review history of their article (what does this mean?). If published, this will include your full peer review and any attached files.

Reviewer #1: No

Reviewer #2: **Yes: **Farhad Shokraneh

---

## [Author Response · Author response to Decision Letter 0]

5 Feb 2021

January, 2nd 2021

Dear editor of PLOS ONE

Ivan D. Florez (Academic Editor)

Thank you for giving us another opportunity to answer the comments made by reviewers in our manuscript entitled: “Use of corticoids and non-steroidal anti-inflammatories in the treatment of rheumatoid arthritis: systematic review and network meta-analysis" (PONE-D-20-24917).

We are looking forward to hearing from you. 

Sincerely,

Dr. Mariana Del Grossi Paglia

University of Sorocaba 

Sorocaba/SP, Brazil

Answer: We check the forms as requested.

2. Thank you for stating the following above the Acknowledgments Section of your manuscript: This project is funded by governmental Program Graduate Education Institutions— PROSUP—CAPES/UNISO. We note that you have provided funding information that is not currently declared in your Funding Statement. However, funding information should not appear in the Acknowledgments section or other areas of your manuscript. We will only publish funding information present in the Funding Statement section of the online submission form.

a. Please remove any funding-related text from the manuscript and let us know how you would like to update your Funding Statement. Currently, your Funding Statement reads as follows: 'The authors received no specific funding for this work.'

c. Please include a separate caption for each figure in your manuscript. 

d. Please include your tables as part of your main manuscript and remove the individual files. Please note that supplementary tables should remain as separate "supporting information" files.

e. Please include captions for your Supporting Information files at the end of your manuscript, and update any in-text citations to match accordingly. Please see our Supporting Information guidelines for more information: http://journals.plos.org/plosone/s/supporting-information

Answers: 

a-b) Thanks for modify the online form. The Brazilian national agency requests that disclosure be made. The information commonly inserted is: “This project is funded by governmental Program Graduate Education Institutions— PROSUP—CAPES/UNISO”.

c) We included a separate caption for each figure in manuscript.

d-e) We included the tables in manuscript file and captions for your supporting information, as requested. 

Additional Editor Comments:

Your manuscript has been reviewed by two experts in the field, and they have found some points that need to be addressed before this manuscript is considered for publication. Additionally, I have performed an Editorial Review. Please go through the reviewers' and Editor's comments, consider addressing these points, and prepare a revised version.

3) Authors did not mention how they assessed and handled the transitivity. This is a major issue as transitivity is one of the core assumptions of NMAs. If not met, the validity of the results is threatened. If this assessment is not present, this should be added as a limitation of the paper.

Answers: Thanks for suggestion. We have corrected this information in the “Method” with the following sentence: “We assess the possibility of intransitivity comparing trials participants between direct comparisons that contributed to an indirect comparisons”.

4) Authors state they applied GRADE NMA approach to assess the quality. However, I see several issues in this application.

• As stated above, how they applied intransitivity, is not clear, is not even mentioned in the paper, and this is one of the GRADE criteria. Puhan's paper, although described the approach, it is not the only guidance, at least 2 more papers have been published by the GRADE group: Brignardello-Petersen et al 2017 and 2018 that should be applied, and they were not In appendix C. There are several issues, such as: why there are not Superscripts in the NMA GRADE assessment? The superscripts used in the indirect assessments show the reasons for downgrading the quality/certainty. in the NMA assessment, according to the GRADE approach, there are at least 2 criteria to be applied (Puhan and Brignardello-Petersen's paper) and they seem to be neglected as none of the NMA has a superscript to describe this.

• again in appendix C, lets take an example with comparison "Placebo v naproxen 1,000 mg". Authors state that in the indirect evidence, there was "inconsistency". A huge problem here, is that inconsistency is evaluated as part of the direct assessment. So this might mean these criteria seems to be the reasons for rating down the direct evidence that informs the indirect estimate. Thus, there is no information to evidence that the Puhan et al. approach was applied, as the criteria for doing that with indirect evidence, is the intransitivity criterion, which, again, was not applied

• As a result of the flaws identified in the GRADE assessment, there is no quality assessment of the NMA estimates, which are the most important ones in NMA. It seems authors only applied the GRADE direct criteria (traditional GRADE assessment), but the indirect and NMA estimates were neglected. Thus, the authors should not describe that they applied the Puhan's approach.

In summary to resolve points described above, you should fully apply the GRADE NMA approach as described by Puhan as you stated (complemented by Brignardello-Petersen's papers 2017 and 2018 (GRADE PAPERS from J of Clin Epidemiol). Two new GRADE approaches by Briganrdello Petersen et al, were published very recently in the BMJ 2020 (PMID: 33177059 and PMID: 33172877), are not mandatory to apply as they were not available when you performed your analysis. IF this is not possible as this time, this should be highlighted as a limitation.

Answers: We inserted the information in “Method” as requested: “We followed the Grading of Recommendations Assessment, Development and Evaluation (GRADE) approach to appraise the confidence in estimates derived from network meta-analysis of outcomes (Brignardello-petersen et al., 2019; Puhan et al., 2014). We corrected the appendix C with the judgment information (superscripts used for NMA assessment). 

5) Table 2 shows the SUCRA values and the mean rank. The mean rank was neither explained in methods, nor in the table. I think the mean rank might be misleading, SUCRA Values (and an explanation, a brief one, in a footnote of the table might be enough). If you want to show the mean, then a variability measure (e.g., SD, 95% CI) should be added.

Answers: We insert the information in “Method” with the phrase: “This parameter was used to estimate the average treatment ranking to explore potential orderings of treatments”. We changed the table 2 by inserting the dispersion measure.

Comments to the Author

i. Is the manuscript technically sound, and do the data support the conclusions? The manuscript must describe a technically sound piece of scientific research with data that supports the conclusions. Experiments must have been conducted rigorously, with appropriate controls, replication, and sample sizes. The conclusions must be drawn appropriately based on the data presented.

Reviewer #1: Partly; Reviewer #2: Partly

ii. Has the statistical analysis been performed appropriately and rigorously?

Reviewer #1: Yes; Reviewer #2: I Don't Know

iii. Have the authors made all data underlying the findings in their manuscript fully available?

Reviewer #1: Yes; Reviewer #2: Yes

iv. Is the manuscript presented in an intelligible fashion and written in standard English?

Reviewer #1: Yes; Reviewer #2: Yes

Reviewer #1:

6) The authors provide a systematic review of NSAIDs and glucocorticoids (GC) in the treatment of rheumatoid arthritis. For the NSAIDs a network meta-analysis (NMA) is done. To establish indirect comparisons in NMA, similarity, homogeneity and consistency must be assessed. In this paper, there is only scarce information about the similarity of the mentioned NSAID studies especially concerning disease activity (as measured by the number of swollen joints, ESR etc.), functional capacity, concomitant DMARD treatment. Although NSAIDs are still used in the treatment of RA, their impact has decreased during the recent years as more potent DMARDs allow for better disease control. Therefore NSAIDs are only used as an adjunct. This might be stressed in the discussion part. Glucocorticoids are mainly used for bridging and their long-term use should be avoided. This should also be mentioned in the discussion part to underline the heterogeneity of the selected GC studies.

Answer: We appreciate your suggestions that contributed to improve the manuscript. The tables 1 and 3 described the characteristics of the studies. We inserted, as requested, information about the functional capacity of disease (in table 1) as well as about the functional capacity and other drugs used by patients (in table 3). We altered the discussion to clarify the adjuvant function of anti-inflammatories and justify the limitations of the studies' findings.

Reviewer #2:

 5) This is a valuable piece of work and admirable that this is done without having any specific funding. I also checked that this is a unique research. There are about 60 other published NMA about RA but none of them have targeted all these medications in one place. I am providing some critical comments that I hope could help the authors and support the editor's decision. I noticed that the authors have not used any of the established RCT search filters that are freely available and are being used as gold standard to retrieve the RCT studies from most of the databases. This seems to be a major issue as the performance of these filters shows that they are the best way of searching RCTs. All these filters are freely available as an appendix for the Search chapter of Cochrane handbook. We usually used Peer-Review of Electronic Search Strategies checklist (PRESS) to peer-review the search at protocol stage. The authors have not followed the items from the PRESS checklist. There are no Controlled Vocabularies used in reported search strategies. In addition, the name of medications/drugs have not been searched at all and the search only contains the class of drugs.

Answer: We thank the comments that contribute to the improvement of this study. We appreciate the PRESS checklist as a good tool for use in searches. It was the first time that we used a filter for this type of study, since normally we did not use it. The RCT filter was necessary because the number of studies found was too large for title and abstract screening. Our search was carried out twice. In the first time, we used the name of all the drugs described in our inclusion criteria, but still there were a very high number of studies identified. We did a new search without including the drug names. Thus, we chose to present this search in the present study since we take care to check if there are the same RCT already selected in the first search.

This month we did a new search strategy using the filters, as recommended. We made a task force to work with a new Endnote spreadsheet. However, of the 26 RCTs included in the manuscript previously, we found only 8 of them, in this new strategy used. In this way, we kept the original strategy, since we think it is more appropriate to results found. In the file entitled “File for Reviewer 2”, you can find this work that we have done. 

6) The authors report that they have searched CENTRAL however they have reported 'Cochrane' in appendix. If by Cochrane they mean Cochrane Library it is different from CENTRAL.

Answer: We made the correction in the supplementary material file (S1 file) to clarify that the research was done in Cochrane (CENTRAL).

7) The authors report that they have searched Web of Science, however they did not mention the names of the databases that have been searched via Web of Science:

Science Citation Index Expanded (SCI-EXPANDED) --1900-present

Social Sciences Citation Index (SSCI) --1956-present

Arts & Humanities Citation Index (A&HCI) --1975-present

Conference Proceedings Citation Index- Science (CPCI-S) --1990-present

Conference Proceedings Citation Index- Social Science & Humanities (CPCI-SSH)-1990-present

Emerging Sources Citation Index (ESCI) -2015-present

Answer: We did not identify access to each database when researched on the Web of Science. As you can see in the figure below, we had the option of searching all databases or searching the most used databases (but the names of the bases were not available). We opted to search in all databases.

8) The authors report that they have searched CINAHL via Ovid SP while since 2006, CINAHL is only available via EBSCOhost.

Answer: We make changes requested.

9) There are discrepancies among PROSPERO protocol, published protocol in Medicine (Baltimore), and the final review submitted to PLoS One in terms of databases and search strategies. The PROSPERO records mentions Scopus as one of the databases to be searched but Scopus is missing from the published protocol and submitted review for PLoS One.

Answer: We performed the search in Virtual Health Library instead of Scopus and we requested this protocol adjustment in PROSPERO. We have included a topic in “Method”: “Differences between protocol and review” in which we explained the changes made in method in relation to the version of the published protocol.

10) Furthermore, the search strategy reported in published protocol in Medicine (Baltimore) is for PubMed. Not only is PubMed not among the searched resources but also the search strategy is different from what reported in the final review for MEDLINE or other databases. Although both search strategies are questionable for not including RCT filters and not following PRESS checklist, one major difference is that the search strategies in the final review do not contain any drug names while the protocol does.

Answer: We changed the “Pubmed” resource and used “the Ovid” resource, since this access allowed us to search other databases. When we demonstrated in the protocol, the strategy “via Pubmed” we intended to provide an example of a search strategy in a database where access is free. As previous mentioned, we searched with two different strategies and I can say that the removal of the names of the drugs did not impact the selection of clinical trials included in our systematic review. The divergences of the published protocol in relation to the version of this manuscript can be verified in the item “Differences between protocol and review”.

11) In published protocol, the authors write in the text that they report search strategy for MEDLINE (via Ovid) in Table 1 and then they report search strategy for MEDLINE (via PubMed). In terms of syntax, the reported search strategy can be run in both interfaces – regardless of its issues – but the discrepancies shows that the search has not been peer-reviewed.

Answer: Thanks for the comment and observation. I don't know if it is possible to make this adjustment in the protocol manuscript, but I will try to contact the journal. When we sent the protocol for publication, we were not aware of how the protocol review would be carried out.

B. Screening

11) In the PRISMA flow diagram, there are 7,615 results after removing the duplicates however the authors only screened 192 records for title and abstract. It is unclear what happens for the remaining 7,423 records. Again in full-text screening, the authors screen 86 full-text and exclude 41 with reason that leaves 45 to include. However, the authors include only 26 not 45. So it is unclear what happens to the remaining 19. I only checked the parts of the manuscript that was related to my area and I leave the rest to the relevant reviewers/editors to decide. Again I want to state that I am impressed with the work done by a determined team for a non-funded project such as this one and I hope the authors find the comments helpful for this review and their future reviews.

Answer: We are grateful for your observation and made the adjustments as requested.

---

## [Decision Letter · Decision Letter 1]

8 Mar 2021

Use of corticoids and non-steroidal anti-inflammatories in the treatment of rheumatoid arthritis: systematic review and network meta-analysis

PONE-D-20-24917R1

Dear Dr. Paglia,

We’re pleased to inform you that your manuscript has been judged scientifically suitable for publication and will be formally accepted for publication once it meets all outstanding technical requirements.

Kind regards,

Ivan D. Florez, MD, MSc, PhD

Academic Editor

PLOS ONE

Additional Editor Comments (optional):

Your revised version have addressed reviewers comments. We are confident this version is an improved one and can be considered for publication

Reviewers' comments:

Reviewer's Responses to Questions

**Comments to the Author**

1. If the authors have adequately addressed your comments raised in a previous round of review and you feel that this manuscript is now acceptable for publication, you may indicate that here to bypass the “Comments to the Author” section, enter your conflict of interest statement in the “Confidential to Editor” section, and submit your "Accept" recommendation.

Reviewer #1: All comments have been addressed

2. Is the manuscript technically sound, and do the data support the conclusions?

Reviewer #1: Yes

3. Has the statistical analysis been performed appropriately and rigorously? 

Reviewer #1: Yes

4. Have the authors made all data underlying the findings in their manuscript fully available?

Reviewer #1: Yes

5. Is the manuscript presented in an intelligible fashion and written in standard English?

Reviewer #1: Yes

6. Review Comments to the Author

Reviewer #1: (No Response)

7. PLOS authors have the option to publish the peer review history of their article (what does this mean?). If published, this will include your full peer review and any attached files.

Reviewer #1: No

---

## [Editor Report · Acceptance letter]

26 Mar 2021

PONE-D-20-24917R1 

Use of corticoids and non-steroidal anti-inflammatories in the treatment of rheumatoid arthritis: systematic review and network meta-analysis 

Dear Dr. Paglia:

I'm pleased to inform you that your manuscript has been deemed suitable for publication in PLOS ONE. Congratulations! Your manuscript is now with our production department. 

Kind regards, 

on behalf of

Dr. Ivan D. Florez 

Academic Editor

PLOS ONE